# Sonodynamic Therapy Using 5-Aminolevulinic Acid for Malignant Gliomas: A Review

**DOI:** 10.3390/life15050718

**Published:** 2025-04-29

**Authors:** Andrea Ebeling, Francesco Prada

**Affiliations:** 1Photonamic GmbH & Co KG, Eggerstedter Weg 12, 25421 Pinneberg, Germany; 2Department of Neurosurgery, Istituto Neurologica Carlo Besta, 20133 Milan, Italy; francesco.prada@istituto-besta.it; 3Department of Neurological Surgery, University of Virginia, Charlottesville, VA 22903, USA; 4Focused Ultrasound Foundation, Charlottesville, VA 22903, USA

**Keywords:** aminolevulinic acid, 5-ALA, focused ultrasound, glioblastoma, protoporphyrin IX, PpIX, sonodynamic therapy, sonosensitizer

## Abstract

In recent years, sonodynamic therapy (SDT) has attracted attention as a promising new approach for the treatment of high-grade gliomas, as it is a non-invasive form of therapy that specifically kills tumor cells with limited side effects. SDT combines low-intensity ultrasound with a sonosensitizer to produce cytotoxic effects in tumor cells. 5-Aminolevulinic acid (5-ALA), an endogenous amino acid that is metabolized to protoporphyrin IX (PpIX), has shown promise as a sonosensitizer for malignant gliomas in SDT. Ultrasound can penetrate deeper body regions and activate PpIX, leading to an increase in tumor immunogenicity and induction of apoptosis. This review highlights the current state of knowledge on the mechanisms of action, the results of preclinical, clinical and ongoing studies on 5-ALA-SDT in malignant gliomas, and discusses the future benefits of SDT.

## 1. Introduction

A new promising treatment approach in malignant gliomas is sonodynamic therapy, which involves the use of ultrasound to generate reactive oxygen species (ROS) from sonosensitizers. Sonodynamic therapy is similar in principle to photodynamic therapy (PDT) but is less invasive and has shown promise in controlling tumor growth in vivo combining 5-ALA with focused ultrasound (FUS) [1]. 5-ALA is a prodrug that is metabolized in malignant gliomas to the fluorescent sensitizer protoporphyrin IX (PpIX). 5-ALA itself is approved for fluorescence-guided resection of high-grade gliomas with sensitization at the approved dose (20 mg/kg body weight); PpIX delivers highly tumor-specific fluorescence in combination with minimal side effects. In the pivotal study [2] and further evaluations [3], the safety concerns regarding drug application or morbidity were no greater than with standard surgery and offered the possibility of safe complete fluorescence-guided resection in glioblastoma (GBM) [4].

PDT with 5-ALA is based on a photochemical reaction induced by the activation of PpIX by laser light of 630 nm in the presence of oxygen, with the generation of reactive oxygen species (ROS) and singlet oxygen molecules, which could ultimately induce apoptosis, autophagic, or necrosis-induced cell death in target tissues [5,6,7,8,9]. Since the light cannot penetrate the skull, PDT requires a craniotomy and direct exposure of the brain tumor in a resection cavity [10,11]; an alternative requires the stereotactic placement of cylindrical diffusing fibers, called interstitial PDT (iPDT), for a precise irradiation of the target tumor volume [12,13]. Despite the ongoing PDT studies on newly diagnosed (NCT03897491) and recurrent high-grade gliomas (NCT04469699), treatment is still limited to small tumor volumes. The main clinical advantage over PDT is that SDT uses the energy of ultrasound waves to activate the sonosensitizer PpIX in producing ROS and trigger cancer cell death [1,14,15,16].

This review provides an insight into the use of focused and diffuse ultrasound in the treatment of GBM using 5-ALA-SDT, its mechanism of action and immune response in preclinical and in vivo models, as well as an overview of current clinical trials and future possibilities as a non-invasive treatment option.

## 2. Sonodynamic Therapy for Malignant Gliomas

High-grade gliomas are the most commonly occurring primary CNS tumors in adults [17]. The current standard-of-care treatment for newly diagnosed HGGs is maximal extent of resection (EOR) for tumors that are surgically accessible followed by radiation therapy and chemotherapy and combined radio- and chemotherapy with median overall survival for newly diagnosed GBM of less than 24 months [18,19]. Today fluorescence-guided surgery with 5-ALA can maximize EOR, which correlates with better survival [4]. Nevertheless, only about half of patients can undergo complete tumor removal due to functional preservation and associated morbidity [20].

Sonodynamic therapy is developing into a promising treatment option for high-grade, inoperable tumors [21]. The principle of SDT is based on the destruction of tumor cells through the sound-induced activation of a sonosensitizing agent that selectively accumulates in neoplastic tissue. While SDT has been shown to be effective in preclinical models for various malignant cancers, including gliomas, there are no drug–device platforms yet that could be used to apply this prospective treatment to patients with this serious disease. The use of focused ultrasound is not new and shows a safe and effective treatment for tremors [22]. An increasing interest in the use of FUS has started in clinical studies for brain cancer demonstrating first SDT efficacy in combination with 5-ALA as a sonosensitizer.

## 3. 5-ALA, an Optimal Sonosensitizer for Malignant Gliomas

5-Aminolevulinic acid (5-ALA) is a precursor of heme synthesis and an endogenous metabolite that is normally formed in the mitochondria from succinyl-CoA and glycine. The metabolization of 5-ALA leads, via a chain of enzymatic reactions, to fluorescent porphyrins, in particular protoporphyrin IX (PpIX), a strong photo- and sonosensitizer, and finally, with the incorporation of iron, to the formation of heme. When 5-ALA is administrated exogenously, it leads to an increased accumulation of PpIX, especially in malignant glioma cells compared to normal cells. Several factors influence the accumulation of PpIX in tumor cells, particularly the reduced activity of ferrochelatase [23], which catalyzes the incorporation of iron into PpIX, resulting in heme. The high accumulation of PpIX in malignant gliomas is being exploited for safer and more complete fluorescence-guided resection by activating PpIX with blue light of 380–420 nm wavelength, resulting in red light fluorescence of 635 nm wavelength in the tumor cells [24].

The positive predictive value of over 98% of the fluorescence evaluation after oral administration of 20 mg/kg body weight of 5-ALA with the strongest fluorescence in the tumor core and at the edges, correlates significantly with the cell density [24,25]. This highly selective activity of PpIX formation in glioblastoma cells may be an advantage for use as a sonosensitizer for SDT treatment of glioblastomas. The oral solution of 5-aminolevulinic hydrochloride (5-ALA HCl) is already approved in Europe and in the US for fluorescence-guided resection (i.e., photodynamic diagnosis) in patients with glioblastoma. 5-ALA HCl has already been used with a safe profile in over 150,000 patients in 40 countries.

### 3.1. 5-ALA-SDT Mechanism of Action

In different in vitro and in vivo studies SDT has proven to be effective although the deep understanding of the mechanism behind 5-ALA-SDT is discussed differently. One of them is based on the so-called sonoluminescence, which involves the conversion of ultrasound into light, resulting in the photoactivation of PpIX through the so-called PDT mechanism which was first demonstrated by Umemura and confirmed by Rosenthal [26,27]. The researchers were able to demonstrate that under acoustic conditions, a sonoluminescence spectrum was generated in the wavelength range from 300 to >500 nm. Sonoluminescence can activate almost the entire absorption spectrum of PpIX. Today, the consensus is that the ultrasound waves during SDT create a cavitation environment in which microbubbles form and that the bursting of the cavitation bubbles releases energy in the form of blue light, which activates tumor cells enriched with PpIX [28]. As a sonosensitizer, the activated PpIX releases singlet oxygen (^1^O_2_) which affects radical oxygen species (ROS) mitochondrial function leading to cell stress which induces the oxidation of pro-apoptotic protein Bcl2 and induces caspase activation [7,29] resulting in the apoptosis of cancer cells confirmed from a morphological and molecular biological perspective [30,31] which can lead to antitumor immunity as shown in Figure 1.

The activation of ROS after SDT was early described [32] and confirmed also in malignant glioblastoma cells (U251) showing that ROS was 61% and 66% greater in the 5-ALA-SDT group compared to the FUS and 5-ALA group alone with induction of apoptosis [33].

### 3.2. 5-ALA-SDT Evidence of Apoptosis, Anti-Tumor Immune Response and Anti-Angiogenesis

Mechanisms of apoptosis induction from SDT have been well characterized. Wang reported that in THP-1, the macrophage-derived foam cells’ intracellular ROS level increased and mitochondrial membrane potential collapsed after 5-ALA-SDT [34]. It was shown that ultrasound also has a synergistic effect together with PpIX and can produce a significant anti-tumor effect. In scanning electron microscopy (SEM), observation shows cellular shrinkage, and the membrane blebbing which also confirmed the apoptotic morphological changes after SDT [32]. Considering that PpIX is synthesized in the mitochondria, the time point at which the highest accumulation of PpIX in the mitochondria occurs may be the best time to induce a burst of apoptosis by 5-ALA-SDT. However, many factors influence the therapeutic effect of SDT, including the intensity, frequency, and duration of the ultrasound.

A significant activation of caspase-3, which plays a central role in the apoptosis pathway, was identified in tumor cells treated with 5-ALA SDT, and an immunofluorescence analysis detected apoptotic features such as the redistribution of Bax/Bak and the release of cytochrome c after SDT, confirming the involvement of a mitochondria-dependent apoptosis pathway [32]. Cytotoxic effects after 5-ALA-SDT led to decreased tumor cell viability, chromatin condensation, and apoptosis in rat RG2 glioma cell lines [28]. 5-ALA with FUS significantly increased caspase-positive cells also in C6 and U87 glioma cell lines [35]. In cells treated with SDT, staining for PARP-1, a mitochondrial protein downstream of the caspase-3 pathway that inhibits DNA repair and drives apoptosis, was significantly increased. Also, Ki67 expression was shown to be significantly decreased in C6 glioma tumors after 5-ALA-SDT [16], supporting that 5-ALA-mediated SDT triggers apoptosis via a mitochondrial-mediated pathway.

Malignant gliomas are primarily infiltrated by M2 microglia/macrophages, which support tumor growth by demanding the immunosuppressive microenvironment. M2 macrophages correlate with a negative prognosis of patients. In mice treated with 5-aminolevulinic acid 5-ALA-SDT, an increased number of M1 CD68+ macrophages could be detected, compared to a significantly reduced number of M2 CD163+ macrophages [34,36,37,38]. There are several ways in which SDT could enhance anti-tumor activity by promoting the conversion of tumor-derived M2-type macrophages to the anti-tumor M1-type. Such influence of 5-ALA-SDT on macrophages and dendritic cells (DCs) was investigated in B16/F10 melanoma tumors [34]. 5-ALA+FUS were able to increase intra-tumoral macrophages. The number of M2-type CD163+ pro-tumorigenic macrophages decreased by SDT. In vitro evidence suggests that 5-ALA-mediated SDT can drive the retraining of pro-tumorigenic macrophages into pro-inflammatory and anti-tumorigenic M1-type (CD86) macrophages. The increase in CD80 and CD86 expression after SDT indicates that DC activation and subsequent antigen presentation to T cells is induced. In addition, the increased expression of the cytokines IL-10, IFN-γ, and TNF-α, which are typically associated with M1 macrophages, shows evidence that SDT promotes the retraining of macrophages to an antitumor phenotype.

It was shown by Galon et al. that the numbers of antigen-specific immune cells within the tumor microenvironment, such as CD8+ T cells, are highly relevant to the clinical prognosis [39]. Since SDT can positively influence the phenotype and number of macrophages and dendritic cells in the tumor microenvironment, it was hypothesized that SDT could most likely influence the positive number and phenotype of tumor-associated immune cells, which in turn could improve treatment outcomes. Peng et al. have discovered for the first time that 5-ALA-SDT has antitumor immunity by inhibiting tumor growth by effectively activating CD8+ T cells and inhibiting the activity of T regulatory cells [40]. It has been shown that 5-ALA-SDT can increase tumor immunogenicity.

Gao et al. reported anti-angiogenic effects of 5-ALA-SDT, where SDT with low ultrasound intensity inhibited the proliferation and migration of endothelial cells in vitro as well as the ability to form capillary networks. Using a rodent model of human tongue cancer, they were able to significantly reduce the expression of vascular endothelial growth factor, an important pro-angiogenic factor, after 5-ALA-SDT treatment [41].

Keenlyside et al. studied SDT on glioma stem cells. The study optimized ultrasound frequency, intensity, plate base material, thermal effect, and live cell integration in a newly developed and validated automated in vitro SDT system. They demonstrated that 5-ALA-SDT induces apoptotic cell death in primary patient-derived glioma cells while upregulating intracellular reactive oxygen species [42].

### 3.3. Preclinical Evidence for the Efficacy of 5-ALA-SDT

In vitro, the first induced cell death in a mouse sarcoma cell line was demonstrated by SDT using hematoporphyrin as a sonosensitizer [26]. First preclinical studies with 5-ALA SDT of rat brains with C6 rat glioma cells achieved a selective antitumour effect against deep-seated experimental glioma with significant smaller tumor size in the SDT group (*p* < 0.05). No harm was exerted to surrounding healthy brain tissue using focused ultrasound (10 W/cm^2^, 1.04 MHz, 5 min); moreover, a much greater reduction in tumor volume in the SDT group compared to controls was achieved [43]. Similar results were obtained by 5-ALA-SDT in C6 glioma models of rats exposed to high-intensity focused ultrasound energy applied directly to the brain surface via craniotomy. 5-ALA-FUS-treated animals showed a significantly slower tumor growth. The control animals did not survive more than 14 days, whereas, in the subgroup of rats that survived to scheduled death, the 5-ALA-SDT cohort showed a more significantly reduced tumor size than the other groups (5-ALA only, and FUS with Radachlorin) (*p* < 0.05). Interestingly, the 5-ALA-SDT group with the 6-h 5-ALA incubation period which corresponds to a high PpIX concentration in the tumor, showed greater inhibition of tumor growth than the 3-h incubation period. Normal brain tissue was not affected by 5-ALA-SDT [14].

Wang et al. published that SDT-induced autophagy decreased at a much higher acoustic intensity (5 W/cm^2^), which might be due to a higher level of cell lysis. They optimized the SDT dose for mild cytotoxicity (1 lg/mL PPIX; 1 W/cm^2^) [32]. 5-ALA-SDT has been tested at different ultrasound operating frequencies. A significantly lower frequency of 25 kHz to treat U87-MG glioma in a nude mice subcutaneous model was used by inducing apoptotic changes of tumor cells [44], while Suehiro has reported that 5-ALA-SDT is promising at a higher frequency of 2.2 MHz in U87 and U251 glioma cells and in U251Oct-3/4 glioma stem cells. They found in in vivo experiments using the U251Oct-3/4 line that 5-ALA-SDT induced both necrosis in the focal area and apoptosis in the perifocal area and reduced the proliferative activity of the entire tumor without damaging the surrounding normal brain tissue. Thereby, the higher-intensity ultrasound with 5-ALA-SDT prolonged the survival in tumor-bearing mice in comparison to the control group without damaging the surrounding normal brain [15]. Yoshida et al. first achieved progress of 5-ALA SDT in malignant gliomas in vitro as well as in vivo. They used the Exablate 4000 Insightecs MRI-guided focused ultrasound (FUS) system (4000 J, 20 W for 240 s), which is approved for high-intensity FUS ablation in the human brain, and found an accumulation of PpIX in the mitochondria of glioma cells leading to the loss of mitochondrial membrane potential, the release of cytochrome c, caspase activation, and apoptosis after sonication [1]. These studies indicate that the SDT effect can be achieved over a wide ultrasound frequency range. In more recent studies by Wu et al. the ultrasound parameters were thus optimized in a C6 rat glioma model; MRgFUS was administered continuously at 1.06 MHz with an in situ spatial-peak temporal-average intensity of 5.5 W/cm^2^ for 20 min [16]. They monitored tumor growth weekly by MRI and were able to show that 5-ALA-SDT significantly improved tumor growth and survival compared to FUS alone and 5-ALA alone. 5-ALA-SDT with relatively low-power continuous-wave FUS can produce an inhibitory effect on glioma growth in the absence of thermal dose.

Treatment regimens and ultrasound parameters were further investigated in new studies with the aim of optimizing them for use in clinical trials. Park et al. investigated the combination of PDT with SDT, which they termed sono-photodynamic phototherapy (SPDT), in a C6 glioma rat model. They found greater tumor shrinkage in the PDT and SPDT groups than with SDT alone, with an increase in cleaved caspase-3 levels being determined after SPDT [45].

Two animal safety studies have been conducted to date. In the first, Raspagliesi et al. investigated the effect of 5-ALA-SDT with low-intensity ultrasound on the healthy brain parenchyma of a large animal porcine model. They used the MRI-guided Insightec ExAblate 4000 220 kHz system. Sonication was performed at different sites in the thalamus and in the periventricular white matter under continuous thermal monitoring. The effects of ultrasound were examined by MRI and histological analysis. The MRI images showed no signs of brain damage, demonstrated safety through lack of edema and hemorrhage and histologic examinations of the areas treated with SDT, and revealed no signs of necrosis or apoptosis attributable to the ultrasound treatments. The swine model confirmed the safety of low-intensity ultrasound combined with 5-ALA. No signs of necrosis or apoptosis were detected in healthy brain tissue following sonication [46].

Further data on the safety and efficacy of 5-ALA-SDT using CV-01-guided low-intensity hemispheric non-ablative ultrasound were evaluated by Pluhar et al. in French bulldogs with naturally occurring high-grade gliomas. The Phase 1 study focused on safety, with secondary outcomes assessing cleaved caspase 3 (CC3) and ionized calcium-binding adaptor molecule 1 (Iba1) as histopathological efficacy markers and overall survival. After one SDT treatment pre-resection, treatments every 3–4 weeks were administered to eight French Bulldogs, totaling 60 administrations (average of 7.5 total treatments/patient). No complications or toxicities were noted, and MRI scans showed no damage to the normal brain. They found canine tumor samples revealed increased apoptosis and immune activation markers post-SDT, e.g., CC3 and Iba1, indicating selective tumor cell death. The median overall survival in the 5-ALA-SDT group reached 205.5 days (6.8 months) compared to 55 days (1.8 months) in controls, with the longest survival at 587 days (versus 213 in controls). The study demonstrated the safety of 5-ALA-mediated SDT in canine glioma patients [47].

Huang et al. evaluated SDT with both sonosensitizers 5-ALA and Sodium Fluorescein in a rat brain tumor model with an advantage for 5-ALA in early-stage brain tumors with relatively intact blood–brain barriers, where 5-ALA penetrates well and promotes the tumor inhibitory effect by SDT, while fluorescein fails to accumulate in the tumor area and no therapeutic effect could be observed. The rats received oral 5-ALA (180 mg/kg body weight) 4 h before FUS treatment (0.4 MPa, duty10%, 20 min). Tumors were significantly reduced in the 5-ALA-SDT group compared to the control, 5-ALA-only and FUS-only groups (control vs. 5-ALA-SDT: *p* = 0.0121, FUS-only vs. 5-ALA-SDT: *p* = 0.0011, 5-ALA-only vs. 5-ALA-SDT: 0.0022). Therapies with FUS-only or 5-ALA-only did not lead to any therapeutic effect on the tumor [48].

Wu et al. stereotactically implanted brain tumor cells in rats and monitored tumor volume by MRI. SDT was repeated weekly with a dose of 60 mg/kg 5-ALA. With a drive frequency of 580 kHz at a pressure of 0.75 MPa, the effect of different burst lengths was investigated to optimize the ultrasound parameters. They also tested 5-ALA-SDT in advanced stage brain tumors. They showed that a longer burst length improved the therapeutic results. With a sonication of 10 ms and 50 ms burst length, the tumor growth could only be reduced in the first three weeks and only with a burst length of 86 ms could the survival result be significantly improved [49].

### 3.4. Current SDT Devices

The use of ultrasound with 5-ALA is being investigated both preclinically and clinically. In contrast to highly focused ultrasound, SDT uses low-intensity focused ultrasound (LIFU) as well as magnetic resonance imaging-guided focused ultrasound (MRgFUS) together with 5-ALA. If the ultrasound application is focused, it requires MRI guidance. Currently, three different devices are used in clinical 5-ALA SDT trials. One is the ExAblate Neuro (InSightec, Tirat Carmel, Israel) device, which is an FUS system approved by the FDA in 2016. In the treatment of essential tremor and Parkinson’s disease, transcranial MRI-guided FUS is performed to deliver 650 kHz pulses [50]. Type 2 of the ExAblate with 220 kHz pulses is being investigated in various clinical studies combined with 5-ALA in newly diagnosed GBM, recurrent GBM, and diffuse intrinsic pontine gliomas (DIPGs). Another device developed and approved by the FDA for the treatment of brain tumors is the NaviFUS (NaviFUS, Taipei City, Taiwan), which uses computed tomography scans to enhance neuronavigation-guided FUS [50]. The technique can disrupt the blood–brain barrier (BBB) to facilitate the passage of drugs. The NaviFUS does not require intraoperative MRI imaging. A study using NaviFUS on 5-ALA-SDT in recurrent GBM (rGBM) is currently under way (NCT06039709). In contrast to the previous devices, the CV01 device (Alpheus Medical, North Oakdale, MN, USA) uses diffuse low-intensity ultrasound. Without MRI, this ultrasound covers the entire hemisphere, which can be useful for GBM, which is a diffusely growing tumor [47]. The method is also called low-intensity diffuse ultrasound (LIDU) [51]. The study using CV01 with 5-ALA in recurrent high-grade gliomas (NCT05362409) has just been completed. Without the MRI guidance, both NaviFUS and CV01 have the advantage for outpatient SDT treatments. Like the NaviFUS is SonoCloud (Carthera, Paris, France). Implanted in the skull, it is another FUS system that delivers LIFUS to disrupt the BBB via microbubbles. It is mainly used in the treatment of brain tumors [50]. This technology was successfully tested in a single-arm phase 1/2 study (NCT03744026) in recurrent GBM and showed 90% efficacy in BBB penetration [52]. Whether this device is suitable for SDT with 5-ALA is currently unknown.

### 3.5. Clinical Trials and Early Evidence of 5-ALA-SDT

The use of SDT as a new treatment option for brain tumors is in its early stages and is being investigated in initial clinical trials as summarized in Table 1. Clinical advantages include that no craniotomy is required and, depending on the device, the procedure can also be performed on an outpatient basis. Initial clinical 5-ALA-SDT studies for brain tumors, including newly diagnosed GBM, rGBM, and DIPG, have shown initial clinical improvements, a lack of observed side effects, and good patient tolerance.

The first-in-human phase 0–1 clinical study (NCT04559685) of 5-ALA-SDT is a single center, open-label study of ascending energy doses of ultrasound under MRI guidance (MRgFUS) using the Exablate 4000 Type 2.0 (220 kHz, Insightec) device to one-half of their tumor volume combined with 5-ALA to assess safety and efficacy in recurrent HGG patients. 5-ALA (10 mg/kg) was administered intravenously 6–7 h prior to FUS treatment. Patients were divided into one of three increasing energy cohorts of MRgFUS (200 J/400 J/800 J, measured at the transducer surface), followed by a four-day interval before tumor resection. The first patient with 5-ALA SDT showed significant tumor shrinkage 4 days after single SDT treatment. In all patients, cleaved caspase-3 (ClCas3) was measured to be elevated as an apoptosis marker in relation to the internal control tissue. Other biomarkers of oxidative stress—4-hydroxynonenal (4HNE), glutathione, cysteine and thiol—were elevated in treated tissue compared to control tissue at all energy levels. For the first time, the same histological biomarkers for cell death as in the SDT animal studies in malignant gliomas were confirmed [53]. The results were presented at the Society for Neuro Oncology (SNO) Annual Meeting 2024 and show that 5-ALA SDT is well tolerated and safe, is not associated with cellular or radiologic off-target effects, and provides direct evidence of reactive oxygen species formation and targeted tumor cell death in recurrent high-grade glioma [54].

In a following phase 1–2 (NCT05370508) multicenter, open-label, dose-escalation and expansion study 5-ALA was administrated iv 6 to 9 h prior to ultrasound under MRI guidance using the Exablate 4000 Type 2.0 device in patients with progressive or relapsed GBM. The safety, dose-limiting toxicities, and the recommended Phase 2 dose for the extension portion of the study were the primary objectives. A 3 + 3 Bayesian design will be used for escalating 5-ALA (5 and 10 mg/kg) and increasing MRgFUS pulse pressure and energy levels (12, 24, and 28 J/subspot) during sonication. It is planned to enroll an additional 36 patients to assess safety and efficacy of repeated cycles of ALA-SDT at the recommended Phase 2 dose (RP2D) dosage. The study protocol was amended for a maximum of 12 treatments at monthly intervals but is currently on hold due to lack of funding [54].

A monocenter, prospective, non-randomized, single-arm phase 2 study (NCT04845919) is using the combination of SDT with MRgFUS (Exablate 4000 Type 2) and the oral administration of 5-ALA HCl 6 h beforehand in patients with newly diagnosed GBM. 5-ALA-SDT treatment is followed by rigorous neuroradiological follow-up (at least 2 MRIs) and tumor resection is performed 15–21 days after SDT, during which tumor tissues are examined for biomarkers of cell death. The primary outcome is represented by the early identification of hemorrhage, edema, or other damages in the first 10 days; secondary outcomes are represented by the evaluation of the rate of neurological deficits and the radiological response to treatment in the first 10 days after the procedure. The study has just been completed, and the results have not yet been published.

The ongoing phase 1 study (NCT06039709) is currently investigating Low-Intensity Focused Ultrasound with the neuronavigation-guided FUS device from NaviFUS Inc (Taipei City, Taipei) in recurrent gliomas. Patients receive oral 5-ALA HCl 6 h beforehand. The one-time 5-ALA-SDT treatment takes place 1–3 weeks before tumor resection, treats 50% of the tumor volume, and serves the remaining 50 percent as internal control. The study is investigating the safety and feasibility as well as histopathological analysis of the resected tissue for signs of cell death.

The phase 1 (NCT05362409) multicenter clinical study evaluated 5-ALA combined with CV01 delivery of LIDU in recurrent high-grade glioma. The study was open to patients with supratentorial tumors only, both grade 3 and 4 astrocytomas as well as grade 3 oligodendrogliomas, with the aims of establishing the recommended duration of CV01 ultrasound transmission and determining the safety and tolerability profile of this intervention; it was sponsored by Alpheus Medical, Inc. (Oakdale MN, USA). The study completed recruitment in March 2023 of originally planned 12 patients for dose-finding with recurrent high-grade gliomas and initial results were presented by Schulder et al. in 2024 [55]. Seven patients were included with multifocal or multicentric disease and two patients after multiple recurrences. 5-ALA-SDT was performed monthly for at least 4 treatments. According to dose escalation and expansion, the treatment duration was set at 120 min. 5-ALA-SDT was well tolerated, with no duration-limiting toxicities or serious associated adverse events. A significant improvement in overall survival (mOS of 14.0+ months, with 7 of 12 patients still alive and final median OS not yet reached) was observed compared to historical data [55].

As MRgFUS allows for real-time thermometric and anatomical information, it was used in a phase 1–2 (NCT05123534) study of 5-ALA-SDT on diffuse intrinsic pontine glioma (DIPG), which is a rare aggressive and deadly paediatric brain tumor [57]. The results were published on the SNO conference by Kilburn et al. and 12 patients with DPIG were treated by SDT combining intravenous 5-ALA (5 mg/kg and 10 mg/kg) with MrgFUS using the Exblate Type 2 device. SDT was performed once per month for a maximum of 12 treatments (range 2–9 treatments). The delivered ultrasonic acoustic energy ranged from 190–444 joules/cm^3^. The median follow-up after 6 months (range 2–18) resulted in 2 partial responses, and in 9 of 12 patients a stable or improved baseline symptom (Lansky Status), including improved mobility and diplopia, could be found. There were no treatment-related adverse events of grade 3 or higher, or dose-limiting toxicities. Two patients died, ten patients are alive a mean of 11 months after diagnosis, and four have discontinued treatment. During treatment, the maximum percentage changes in tumor volume ranged from a reduction of −68% to an increase of over 100% [56].

Initial results are available from a case study published by Stummer combining 5-ALA with low-intensity, non-targeted ultrasound (LIFU) as neo-adjuvant treatment in three patients of naïve newly diagnosed GBM. 5-ALA HCl at the approved dose for fluorescence-guided resection (20 mg/kg body weight) was administered orally 6 h prior to sonication with the CV01 device (Alpheus Medical, North Oakdale, MN, USA) [51]. After shaving the hair, the ultrasound transducer was successively placed on the hemisphere at 10 different treatment sites for sonication of the entire hemisphere. The ultrasound was emitted with a sound intensity of 2–10 W/cm² and a frequency of 0.750–1.1 MHz. The ultrasonic emission comprises a pulsed method with a low duty cycle, which is based on an Isppa of 10 W/cm^2^ (~550 kPa) and a mechanical index (MI) of about 0.6 in situ (in the diseased tissue). The total treatment time per site was 12 min. The skull thickness at each of the 10 treatment sites was pre-assessed using CT imaging, and the ultrasound power was scaled according to the skull thickness and expected transmission losses to the target value for Isppa of 10 W/cm^2^ in situ. MRI was performed 24–48 h after 5-ALA-SDT. During resection, which took place 3–4 days after SDT, tissue samples were taken from the peripheral invasive tumor zone of each patient for neuropathological and immunohistochemical analysis. No adverse events were noted in all three patients after a single 5-ALA-SDT. An MRI scan less than 48 h after SDT showed no gross increase in edema or tumor mass. As shown in Figure 2, the diffusion-weighted imaging showed a marked increase in signal.

As shown in Figure 3, the patients with evaluable perfusion studies before and after sonography (patients 2 and 3) showed a measurable increase in leakage and perfusion in both tumors.

The apparent diffusion coefficient (ADC) gave a corresponding decrease in signal, which was confined to the tumor margins and consequently quantified. Patients showed a measurable increase in leakage and perfusion in both tumors. Thus, the decrease in apparent diffusion coefficient values in tumors in the post-MRI indicated a cytotoxic effect. After 5-ALA-SDT treatment, a high expression of the apoptosis marker cleaved caspase 3 were detected in all patients, and Figure 4 shows the expression of cleaved caspase 3 in one patient 7 days before sonication and after surgery, and in 2 patients after surgery.

The authors concluded that after a single 5-ALA-SDT, an immediate marked imaging response indicating cytotoxic edema and evidence of histopathologic response could be measured.

## 4. Conclusions and Future Direction of the 5-ALA-SDT

Despite maximal tumor resection in glioblastoma followed by radio- and/or chemotherapy, recurrences occur within months and there is no standard for their therapy [58]. Therapies that prolong survival are urgently needed, and SDT could be a new approach. In addition, the penetration of ultrasound could also reach brain regions where tumor resection is not feasible such as the infiltration zone near functionally important brain regions. Compared to photodynamic therapy, where the sensitizer is activated with light, ultrasound reaches the sonosensitizers even in deep tumor areas [59]. In different animal models, it has been shown that 5-ALA-SDT reduced the growth of gliomas with a positive effect on prolonged survival. The therapeutic efficacy also showed a proportionality to the burst length where longer burst lengths during ultrasound exposure enhanced therapeutic outcomes, suggesting the importance of sustained interaction between ultrasound and the sonosensitizer [49]. In comparison to a single SDT, the repetition of 5-ALA-SDT resulted in significant antitumor effects in preclinical trials, and longer survival.

There are currently no approved drug–device platforms to administer 5-ALA-SDT to patients with high-grade gliomas. The currently ongoing clinical trials with the combination product 5-ALA HCl oral solution and the different ultrasound devices (Exblate Type 2, NaviFUS and CV01) for delivering low-intensity ultrasonication to deep brain regions and inducing apoptosis of PpIX-containing tumor cells in newly diagnosed and recurrent glioblastoma are important to obtain initial experience on safety and efficacy. Crossing the blood–brain barrier remains a critical challenge in selecting the appropriate sonosensitizer, and here 5-ALA has proven to be ideal as it is an approved drug with a recommended oral dosage of 20 mg/kg body weight used for fluorescence-guided resection of high-grade gliomas and is used in photodynamic therapy [2,10]. Instead, low-grade gliomas usually have an intact or only slightly disrupted BBB. Since 5-ALA cannot efficiently overcome an intact BBB, it does not achieve sufficient accumulation in low-grade glioma tissue to allow effective SDT [60], in contrast to high-grade gliomas where the BBB is disrupted. However, temporary opening of the BBB by microbubbles using focused ultrasound may be a promising non-invasive technique to improve the therapeutic delivery of 5-ALA to low-grade gliomas for SDT treatment. MB-FUS is safe, controllable, and has been shown to be effective in both preclinical and early clinical phases [50,61,62].

Unlike most primary brain tumors, meningiomas are located outside the BBB, and PpIX fluorescence induced by 5-ALA is also currently used in the resection of all grade I-III meningiomas [63,64]. SDT has not yet been tested as a treatment option but could be an alternative to radiotherapy in the future.

To date, no edema formation has been reported after 5-ALA-SDT as seen after 5-ALA-PDT [65,66], and as a non-invasive treatment, SDT has the chance to be repeated compared to PDT. The risk of potential thermal damage to tissue from low-intensity ultrasound waves such as those used in SDT, is anticipated to be very low, especially when a cooling system is integrated into the device.

In initial clinical trials of SDT in GBM, no serious adverse events (SAEs) were found [51,53,54,57]. Since the studies published to date are still in early phases, the results of the ongoing studies are much awaited to evaluate safety and efficacy, and the influence of a single 5-ALA-SDT versus a repeat SDT treatment needs to be considered. In recurrent high-grade glioma (NCT05362409), 5-ALA SDT showed a doubling of median overall survival (15.7 vs. 6–8 months historically) and a tripling of progression-free survival (5.5 vs. 1.8 months) [55]. In the first six patients from the DIPG study (NCT05123534), two achieved a partial response as determined by a centralized review using the RANO criteria. Two patients continued treatment and were progression-free 11 to 15 months after starting SDT, which is remarkable given the median survival time of 9–12 months typical for DIPG [56]. Preliminary efficacy in HGG and DIPG is promising, with partial response and prolonged progression-free survival. Safety shows no grade ≥ 3 adverse events associated with SDT treatment. In addition, SDT was well tolerated and the studies allowed one monthly repeat per patient.

The early initial results of sonodynamic treatment before surgery may indicate a possibility as a neoadjuvant treatment [51]. As a non-invasive treatment, it also offers the possibility of an adjuvant treatment that could be used before or possibly instead of radio-/chemotherapy. Repeated SDT could thereby improve quality of life of patients, especially in children, compared to conventional standard forms of therapy.

SDT may also be considered as a primary treatment strategy for patients who are not suitable candidates for surgery or who have recurrent tumors that are difficult to treat with other modalities. The use of SDT as a treatment for glioblastoma needs to be investigated in further clinical trials. SDT is capable of treating diffusely growing tumors even in deeper areas of the skull in a non-invasive manner [51]. For larger tumors, however, the challenge is to capture the entire tumor, including the infiltrating margins, with optimal ultrasound parameters. In addition, the effectiveness of SDT is closely related to the total energy delivered, which depends on factors such as burst length and ultrasound frequency. Optimizing these parameters for larger tumor volumes can be a technical challenge depending on the device. In particular, high-grade gliomas are characterized by a large heterogeneity that affects treatment prognosis, and further studies are needed to clarify what advantage SDT might have for a better prognosis, e.g., for patients with MGMT-positive or -negative status. In addition, gliomas inherently have a heterogeneous microenvironment, including areas of hypoxia or necrosis, which may reduce the metabolization of 5-ALA and the activation of PPIX as a sonosensitizer, potentially limiting the efficacy of SDT in some tumor regions. Initial studies have shown that repeated SDT, when carefully planned to cover the entire tumor volume, has significant antitumor effects and even resulted in complete tumor eradication in some preclinical studies. Therefore, it remains important to optimize SDT such as longer bursts and weekly or monthly treatment regimen in future studies.

Gliomas are a heterogeneous group of tumors with distinct histological and molecular subtypes, such as IDH-wildtype glioblastoma, IDH-mutant astrocytoma, and oligodendroglioma. Most preclinical and clinical SDT studies have focused on glioblastoma multiforme (GBM), particularly due to its aggressiveness and poor prognosis. However, the molecular and microenvironmental differences between glioma subtypes—such as variations in immune cell infiltration, gene expression, and tumor microenvironment—may influence SDT efficacy [67]. For instance, studies show that glioma subtypes exhibit distinct immune microenvironments, with differences in macrophage, neutrophil, and T cell infiltration, which could impact the response to SDT-induced reactive oxygen species (ROS) and immune modulation. IDH-mutant gliomas, which tend to be less aggressive, may respond differently to SDT compared to the highly heterogeneous and immunosuppressive IDH-wildtype GBMs [68]. Therefore, the scalability of SDT across different glioma subtypes requires tailored treatment approaches that address these biological differences.

In addition to glioblastomas, SDT has been preclinically tested in breast, pancreatic, or liver cancer using PpIX as sonosensitizer [69,70,71]. Foglietta et al. tested 5-ALA-SDT in a breast cancer rat model, where the SDT-treated group showed a significant reduction (*p* ≤ 0.0001) in tumor size compared to the untreated group and the groups treated with 5-ALA and ultrasound alone, and a strong decrease (*p* ≤ 0.0001) in Ki67 protein expression was the most important feature of the SDT-treated cancer tissues. Oxidative stress could be confirmed as a crucial driver of the anticancer effect through cell cycle arrest, apoptosis, and autophagy [69]. Thus, 5-ALA-SDT could also be conceivable as a new therapeutic strategy for the treatment of a variety of cancer types.

Another possibility of SDT is its combination with chemotherapeutic drugs. Osaki combined 5-ALA-SDT with bleomycin in vitro and in vivo and results suggest that the mechanism of tumor shrinkage induced by 5-ALA-SDT with the chemotherapeutic drug may involve not only direct cell killing but also vascular blockage and enhancement of the efficacy of bleomycin [72]. Recent publications have investigated temozolomide (TMZ), a DNA-alkylating drug that prolongs overall survival in patients with GBM, in combination with ultrasound (US). Wang showed that TMZ can generate reactive ROS under the influence of US on glioma cells in vitro and in vivo in a mouse model [73]. Zhou et al. found that SDT with TMZ as a sonosensitizer disrupted endoplasmic reticulum and mitochondrial morphology and normal functions, and increased mitochondrial permeability indicated that intracellular protein synthesis and energy metabolism were impaired. In addition to DNA alkylation by TMZ, the division of malignant glioma cells was severely restricted by SDT [74]. With PpIX as a sonosensitizer, the accumulation of more intracellular ROS in mitochondria may damage them and release cytochrome c and apoptosis-inducing factors, leading to mitochondria-dependent apoptosis. A combination of TMZ and PpIX as a sonosensitizer could therefore be another approach for targeted SDT treatment of high-grade gliomas with ultrasound.

There is growing interest in combining SDT with established approaches such as chemotherapy and immunotherapy to develop more comprehensive and effective strategies for malignant glioma. This could overcome drug resistance, improve tumor cell killing through synergistic mechanisms, and reduce toxicity through lower doses of individual agents. While preclinical and early clinical data are encouraging, further research is needed to optimize these strategies, understand their toxicity profiles, and identify the patients most likely to benefit.

Wang has shown that in addition to producing tumor antigens, SDT can also reduce immune tolerance, thereby reversing immune tolerance to tumor antigens and stimulating antitumor immune effects [75]. They combined an immune-adjuvant-containing nano-sonosensitizer with anti-PD-L1 checkpoint inhibitor, which inhibited the growth of primary tumors and induced an effective systemic immune response. In addition, the combination therapy resulted in long-term immunological memory and prevented tumor recurrences [76,77]. SDT combined with immunotherapy was also shown to be more effective in inhibiting tumor growth and metastasis by enhancing antitumor immunity [78]. However, studies using PpIX as a sonosensitizer have not yet been conducted.

SDT induces cytotoxicity primarily through ROS, which can damage both tumor cells and, to a lesser extent, healthy cells. Combining SDT with PDT could achieve synergistic effects, increasing ROS production and thus potentially enhancing cytotoxicity, but also increasing the risk of damage to surrounding healthy tissue if not targeted.

SDT combined with chemotherapeutic agents could increase the risk of overlapping toxicities (e.g., increased fatigue, nausea, or organ-specific side effects) and should be monitored, especially if the chemotherapeutic agent has known side effects on the same organ system targeted by SDT. Combining SDT with radiation could theoretically increase local tissue inflammation or damage through additive ROS generation, but clinical data are limited. Careful patient selection and monitoring would be required to avoid exacerbation of radiation-induced side effects, especially in sensitive organs. There is limited direct evidence, but any therapy that increases ROS or inflammatory responses could potentially increase immune-related side effects or organ toxicity. The combination could also have beneficial effects by enhancing immune recognition of tumor cells. However, this must be balanced against the risk of increased toxicity. Therefore, it will be necessary to evaluate the safety profile of SDT, especially in combination therapies.

The selection of the ideal sonosensitizer in SDT is crucial and should have high tumor cell affinity with minimal impact on healthy brain parenchyma. While primarily used for fluorescence-guided resection of gliomas, 5-ALA has demonstrated potential beyond surgical applications, particularly in non-invasive SDT. Its use during surgery improves complete resection rates and progression-free survival, which may complement SDT for residual or inaccessible tumors, by using low-frequency ultrasound, which can penetrate the human skull safely. Nevertheless, further clinical work is necessary for 5-ALA-SDT as a new treatment option of high-grade gliomas. The optimization of ultrasound parameters (e.g., frequency and burst length) is critical for maximizing therapeutic outcomes while minimizing side effects, as is the repetition frequency of 5-ALA-SDT depending on tumor respone. While SDT shows great promise, it is important to note that clinical trials are still ongoing, and larger clinical trials are needed to validate preclinical findings and establish standardized protocols for 5-ALA-mediated SDT. It may be particularly useful for treating residual or inaccessible tumors after surgical resection. However, the preclinical results and early clinical investigations suggest that SDT could become a valuable addition to the treatment arsenal for malignant gliomas, potentially offering improved outcomes with fewer side effects compared to traditional therapies. There may also be a promising future for the combination of SDT with other treatments, such as chemotherapy or immunotherapy, to enhance overall efficacy.

In summary, 5-ALA is a highly effective sonosensitizer for malignant glioma, offering significant therapeutic potential when combined with focused ultrasound. Its ability to selectively target tumor cells while sparing healthy tissue positions it as a promising tool in the non-invasive treatment of gliomas, but more clinical studies are required to translate these findings into standard care.

## Figures and Tables

**Figure 1 life-15-00718-f001:**
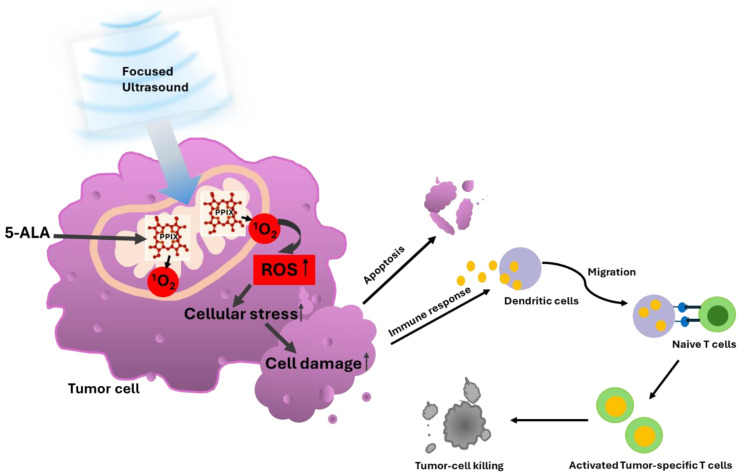
The figure represents the 5-ALA-SDT effect. After systemic administration of 5-ALA, it metabolized into PPIX, which accumulates mainly in the mitochondria of tumor cells. Upon activation by low-intensity ultrasound, PpIX generates oxygen species (ROS), leading to an increase in cellular stress in the surrounding mitochondria, cell membrane, and other organelles, resulting in tumor cell damage that triggers apoptosis and an antitumor immune response.

**Figure 2 life-15-00718-f002:**
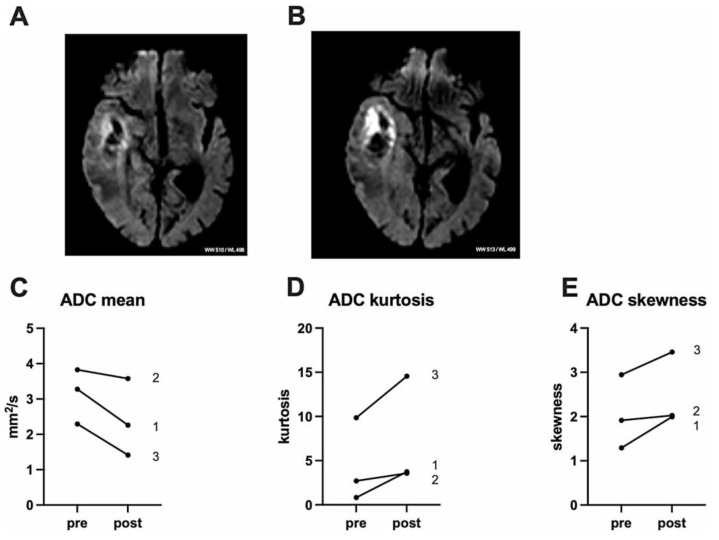
The figure represents the illustrative example of diffusion imaging of patient 1 before (**A**) and after (**B**) sonication (TR: 2658.3 ms; TE: 80.8 ms, ST: 6 mm/SP: 7 M). Change in diffusion imaging values ADC mean, ADC kurtosis, and ADC skewness in all patients, labeling refers to patient number ((**C**–**E**), respectively). Paired *t*-testing resulted in *p* 0.09 (**C**), *p* 0.12 (**D**), and *p* 0.13 (**E**). ADC, apparent diffusion coefficient [51]. Copyright by CC BY 4.0 https://creativecommons.org/licenses/by-nc/4.0/deed.en (accessed on 19 March 2025).

**Figure 3 life-15-00718-f003:**
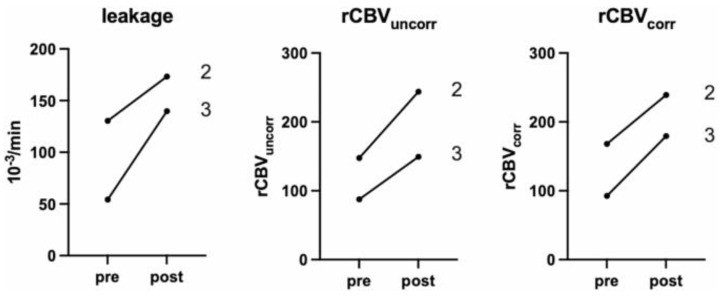
The figure represents the change in tumor perfusion. Labeling refers to patient number. Paired *t*-testing resulted in *p* 0.12 (leakage), *p* 0.14 (rCBVuncorr), and *p* 0.06 (rCBVcorr). rCBVuncorr, uncorrected relative cerebral blood volume; rCBVcorr, corrected relative cerebral blood volume [51]. Copyright by CC BY 4.0 https://creativecommons.org/licenses/by/4.0/deed.en (accessed on 19 March 2025).

**Figure 4 life-15-00718-f004:**
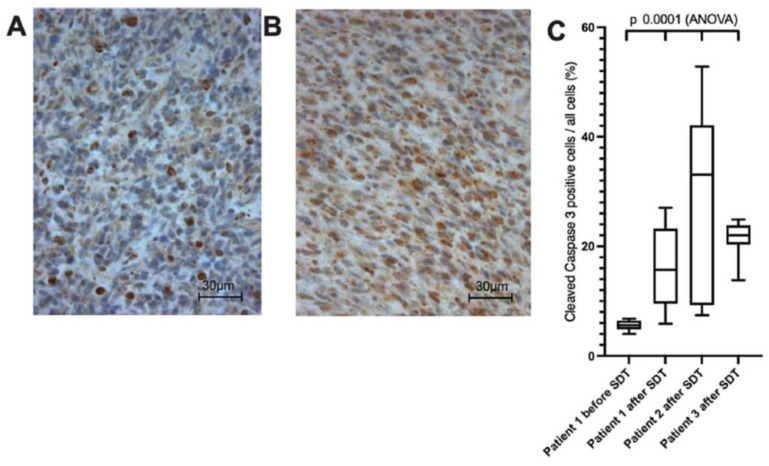
The figure represents anti cleaved caspase 3 immunohistochemistry in patient 1 before (**A**) and after (**B**) SDT. Comprehensive tissue analysis of all patients (**C**) [51]. Copyright by CC BY 4.0 https://creativecommons.org/licenses/by/4.0/deed.en (accessed on 19 March 2025).

**Table 1 life-15-00718-t001:** The table represents an overview of clinical trials on 5-ALA-SDT in malignant gliomas.

Phase Trial Name	Identifier	Device	Status/ Published
0–1 Study of SDT Therapy in Participants with Recurrent High-Grade Glioma (HGG)	NCT04559685	Exablate 4000 Type-2 Device (220 kHz)	Ongoing [53,54]
1–2 A Study of Sonodynamic Therapy Using SONALA-001 and Exablate 4000 Type 2.0 in Subjects With Recurrent GBM	NCT05370508	Exablate 4000 Type-2 Device (220 kHz)	Study is terminated due to funding challenges and not due to safety concerns [54]
1 Sonodynamic Therapy in Patients With Recurrent GBM (GBM 001)	NCT06039709	NaviFUS	Ongoing N/A
1 Study to Evaluate 5-ALA Combined With CV01 Delivery of Ultrasound in Recurrent High- Grade Glioma	NCT05362409	Alpheus	Complete [55]
2 Sonodynamic Therapy with ExAblate System in Glioblastoma Patients (Sonic ALA)	NCT04845919	Exablate 4000 Type-2 Device (220 kHz)	Complete N/A
1–2 A Phase 2 Study of Sonodynamic Therapy (SDT) Using SONALA-001 and ExAblate 4000 Type 2.0 in Patients with Diffuse Intrinsic Pontine Glioma (DIPG)	NCT05123534	Exablate 4000 Type-2 Device (220 kHz)	Suspended due to lack of funding but not due to safety concerns [56]

N/A—not applicable.

## Data Availability

The data analyzed in the current review are accessible from the corresponding references.

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
