# Peer review of "Sonodynamic Therapy Using 5-Aminolevulinic Acid for Malignant Gliomas: A Review"

_life, 2025, doi:10.3390/life15050718_

Round 1
Reviewer 1 Report
Comments and Suggestions for Authors
The manuscript is a valuable addition to the field and provides substantial insights into the potential of SDT in glioma treatment. It is well-structured, and comprehensive in its coverage of both preclinical and clinical data. However, further critical analysis is needed in some areas, particularly regarding the clinical applicability of SDT and its scalability. Therefore, I recommend revision in following aspects:
- While the manuscript discusses the promising results of SDT in preclinical studies, a more thorough analysis of the challenges faced in clinical translation is needed. The authors should elaborate on the difficulties in scaling SDT for larger or more diffuse tumors, as well as address the issues related to the heterogeneous nature of gliomas.
- The potential limitations of SDT in terms of tissue penetration, particularly in solid tumors, and its ability to target heterogeneous tumor populations need to be discussed. Moreover, the manuscript would benefit from a discussion on the role of the blood-brain barrier (BBB) and how SDT could overcome this challenge in glioma treatment.
- The authors should provide a more detailed account of the long-term safety and efficacy outcomes from clinical trials. While the manuscript mentions several clinical trials, a deeper exploration of the outcomes, including patient survival rates and the comparison of SDT with other established therapies, would strengthen the manuscript.
- A more in-depth discussion on the potential for combining SDT with other treatment modalities, such as chemotherapy or immunotherapy, would be beneficial. This could address the clinical need for more comprehensive therapeutic strategies, especially for aggressive glioma subtypes.
- There should be a more detailed exploration of the scalability of SDT for different glioma subtypes and tumor sizes. The current focus on a particular type of glioma should be expanded to consider the broader applicability of SDT across other malignancies.
- While the manuscript touches on safety issues, it would be beneficial to further expand this section. Addressing any potential side effects of SDT, especially when used in combination with other therapies, would provide a more balanced view of its clinical potential.
Author Response
1. While the manuscript discusses the promising results of SDT in preclinical studies, a more thorough analysis of the challenges faced in clinical translation is needed. The authors should elaborate on the difficulties in scaling SDT for larger or more diffuse tumors, as well as address the issues related to the heterogeneous nature of gliomas.
Thank you for pointing this out. We have added on page 13, pragraph 4, line 502-518 the following: SDT is capable of treating diffusely growing tumors even in deeper areas of the skull in a non-invasive manner [51]. For larger solid tumors, however, the challenge is to capture the entire tumor, including the infiltrating margins, with optimal ultrasound parameters. In addition, the effectiveness of SDT is closely related to the total energy delivered, which depends on factors such as burst length and ultrasound frequency. Optimizing these parameters for larger tumor volumes can be a technical challenge depending on the device. In particular, high-grade gliomas are characterized by a large heterogeneity that affects treatment prognosis, and further studies are needed to clarify what advantage SDT might have for a better prognosis e.g. for patients with MGMT-positive or -negative status. In addition, gliomas inherently have a heterogeneous microenvironment, including areas of hypoxia or necrosis, which may reduce the metabolization of 5-ALA and the activation of PPIX as a sonosensitizer, potentially limiting the efficacy of SDT in some tumor regions. Initial studies have shown that repeated SDT, when carefully planned to cover the entire tumor volume, has significant antitumor effects and even resulted in complete tumor eradication in some preclinical studies [49]. Therefore, it remains important to optimize SDT such as longer bursts and weekly or monthly treatment regimen in future studies.
2. The potential limitations of SDT in terms of tissue penetration, particularly in solid tumors, and its ability to target heterogeneous tumor populations need to be discussed. Moreover, the manuscript would benefit from a discussion on the role of the blood-brain barrier (BBB) and how SDT could overcome this challenge in glioma treatment.
We agree that this additional information is good to include. We also discuss in more detail under page 13, paragraph 4 on your first question the heterogeneous situation of the solid gliomas. In addition we put on page 12,paragraph 4, line 463 - 474 the following information regarding overcoming BBB: Instead, low-grade gliomas usually have an intact or only slightly disrupted BBB. Since 5-ALA cannot efficiently overcome an intact BBB, it does not achieve sufficient accumulation in low-grade glioma tissue to allow effective SDT [72], in contrast to high-grade gliomas where the BBB is disrupted. However, temporary opening of the BBB by microbubbles using focused ultrasound may be a promising non-invasive technique to improve the therapeutic delivery of 5-ALA to low-grade gliomas for SDT treatment. MB-FUS is safe, controllable and has been shown to be effective in both preclinical and early clinical phases [50,73-74]. Unlike most primary brain tumors, meningiomas are located outside the BBB, and PpIX fluorescence induced by 5-ALA is also currently used in the resection of all grade I-III meningiomas [75-76]. SDT has not yet been tested as a treatment option but could be an alternative to radiotherapy in the future.
3. The authors should provide a more detailed account of the long-term safety and efficacy outcomes from clinical trials. While the manuscript mentions several clinical trials, a deeper exploration of the outcomes, including patient survival rates and the comparison of SDT with other established therapies, would strengthen the manuscript.
Thank you for pointing this out. In paragraph 3.5. we explained safety and efficacy data so far published for all clinical studies, most of them ongoing and long-term data still awaiting but we agree to include on page 12-13, paragraph 4, line 483 - 493 a more detailed look: In recurrent high-grade glioma (NCT05362409), 5-ALA SDT showed a doubling of median overall survival (15.7 vs. 6-8 months historically) and a tripling of progression-free survival (5.5 vs. 1.8 months) [55]. In the first six patients from the DIPG study (NCT05123534), two achieved a partial response as determined by a centralized review using the RANO criteria. Two patients continued treatment and were progression-free 11 to 15 months after starting SDT, which is remarkable given the median survival time of 9-12 months typical for DIPG [56]. Preliminary efficacy in HGG and DIPG is promising, with partial response and prolonged progression-free survival. Safety shows no grade ≥ 3 adverse events associated with SDT treatment. In addition, SDT was well tolerated and the studies allowed one monthly repeat per patient.
4. A more in-depth discussion on the potential for combining SDT with other treatment modalities, such as chemotherapy or immunotherapy, would be beneficial. This could address the clinical need for more comprehensive therapeutic strategies, especially for aggressive glioma subtypes.
Thank you for pointing this out and we agree to include on page 14, paragraph 4, line 559-565 a better potential overview: There is growing interest in combining SDT with established approaches such as chemotherapy and immunotherapy to develop more comprehensive and effective strategies for malignant glioma. This could overcome drug resistance, improve tumor cell killing through synergistic mechanisms, and reduce toxicity through lower doses of individual agents. While preclinical and early clinical data are encouraging, further research is needed to optimize these strategies, understand their toxicity profiles, and identify the patients most likely to benefit.
5. There should be a more detailed exploration of the scalability of SDT for different glioma subtypes and tumor sizes. The current focus on a particular type of glioma should be expanded to consider the broader applicability of SDT across other malignancies.
Thank you for pointing this out and we agree to include on page 13, paragraph 4, line 519- 532 a better overview: Gliomas are a heterogeneous group of tumors with distinct histological and molecular subtypes, such as IDH-wildtype glioblastoma, IDH-mutant astrocytoma, and oligodendroglioma. Most preclinical and clinical SDT studies have focused on glioblastoma multiforme (GBM), particularly due to its aggressiveness and poor prognosis. However, the molecular and microenvironmental differences between glioma subtypes—such as variations in immune cell infiltration, gene expression, and tumor microenvironment—may influence SDT efficacy [77]. For instance, studies show that glioma subtypes exhibit distinct immune microenvironments, with differences in macrophage, neutrophil, and T cell infiltration, which could impact the response to SDT-induced reactive oxygen species (ROS) and immune modulation. IDH-mutant gliomas, which tend to be less aggressive, may respond differently to SDT compared to the highly heterogeneous and immunosuppressive IDH-wildtype GBMs [78]. Therefore, the scalability of SDT across different glioma subtypes requires tailored treatment approaches that address these biological differences.
6. While the manuscript touches on safety issues, it would be beneficial to further expand this section. Addressing any potential side effects of SDT, especially when used in combination with other therapies, would provide a more balanced view of its clinical potential.
Thank you for pointing this out and we agree to include on page 14, paragraph, 4 line 575-590 a better potential overview: SDT induces cytotoxicity primarily through ROS, which can damage both tumor cells and, to a lesser extent, healthy cells. Combining SDT with PDT could achieve synergistic effects, increase ROS production and thus potentially enhancing cytotoxicity, but also increasing the risk of damage to surrounding healthy tissue if not targeted. SDT combined with chemotherapeutic agents could increase the risk of overlapping toxicities (e.g., increased fatigue, nausea, or organ-specific side effects) and should be monitored, especially if the chemotherapeutic agent has known side effects on the same organ system targeted by SDT. Combining SDT with radiation could theoretically increase local tissue inflammation or damage through additive ROS generation, but clinical data are limited. Careful patient selection and monitoring would be required to avoid exacerbation of radiation-induced side effects, especially in sensitive organs. There is limited direct evidence, but any therapy that increases ROS or inflammatory responses could potentially increase immune-related side effects or organ toxicity. The combination could also have beneficial effects by enhancing immune recognition of tumor cells. However, this must be balanced against the risk of increased toxicity. Therefore, it will be necessary to evaluate the safety profile of SDT, especially in combination therapies.
Reviewer 2 Report
Comments and Suggestions for Authors
In this work, Ebeling and Prada present a literature review about sonodynamic therapy (SDT). More specifically, it discusses using aminolevulinic acid (5-ALA) as a sonosensitizer for malignant gliomas. This work highlights the current state-of-the-art regarding mechanisms of action and results of ongoing preclinical and clinical studies on 5-ALA-SDT in malignant gliomas. This is an interesting study that could merit publication in this journal, but some issues should be addressed:
1-The authors focus on malignant gliomas. Why this specific focus?
2-The authors identify 5-ALA as a sonosensitizer, but if PpIX is activated by ultrasounds, isn't the latter compound the actual sonosensitizer?
3-What other sonosensitizers exist and have been studied?
4-The authors should indicate if other reviews have already been published on this topic and, if so, what is the novelty of the present work.
Author Response
1-The authors focus on malignant gliomas. Why this specific focus?
We agree that additional information to malignant gliomas would be good to include and to inform here also about the role of the BBB. Therefore, we put on page 12, paragraph 4, line 463 - 474 the following text: Instead, low-grade gliomas usually have an intact or only slightly disrupted BBB. Since 5-ALA cannot efficiently overcome an intact BBB, it does not achieve sufficient accumulation in low-grade glioma tissue to allow effective SDT [72], in contrast to high-grade gliomas where the BBB is disrupted. However, temporary opening of the BBB by microbubbles using focused ultrasound may be a promising non-invasive technique to improve the therapeutic delivery of 5-ALA to low-grade gliomas for SDT treatment. MB-FUS is safe, controllable and has been shown to be effective in both preclinical and early clinical phases [50,73-74]. Unlike most primary brain tumors, meningiomas are located outside the BBB, and PpIX fluorescence induced by 5-ALA is also currently used in the resection of all grade I-III meningiomas [75-76]. SDT has not yet been tested as a treatment option but could be an alternative to radiotherapy in the future.
2-The authors identify 5-ALA as a sonosensitizer, but if PpIX is activated by ultrasounds, isn't the latter compound the actual sonosensitizer?
Thank you for pointing this out and you are right, therefore, we wrote on page 1, abstract, line 14-16 that “5-Aminolevulinic acid (5-ALA), an endogenous amino acid that is metabolized to protoporphyrin IX (PpIX), has shown promise as a sonosensitizer for malignant gliomas in SDT.” Also, on page 2, paragraph 1, line 47-48 it is written that” …SDT uses the energy of ultrasound waves to activate the sonosensitizer PpIX in producing ROS and trigger cancer cell death...” Therefore, we explained it in different parts of the manuscript that PPIX and not 5-ALA is the sonosensitizer.
3-What other sonosensitizers exist and have been studied?
Thank you for pointing this out. Since this review focuses exclusively on 5-ALA and its sonosensitizer PPIX, we have only mentioned TMZ as a possible sonosensitizer on page 14, paragraph 4, lines 549–550, and an immunoadjuvant-containing nano-sonosensitizer on page 14, paragraph 4, lines 568–570. However, we believe that this review should not provide an overview of other sonosensitizers and should focus on PPIX.
4-The authors should indicate if other reviews have already been published on this topic and, if so, what is the novelty of the present work.
Thank you for pointing this out. A previous review by Marcus, cited in our manuscript in [61], describes in detail 5-ALA photodynamic therapy (PDT) in various indications, discusses fluorescence, and then describes SDT with 5-ALA. Our review addresses SDT in brain tumors, focusing on the latest preclinical study results as well as the most recent clinical trial data. We also provide information on the latest patient results with 5-ALA SDT, referring to the data presented in section 3.5 of this publication and also cited in Stummer [51], which underscores the novelty of this review. The outlook in paragraph 4 with the recent reference to possible combinations with TMZ, immunotherapy and the outlook for brain and possibly other solid tumors as well as the newly added critical discussion of combination therapies and the heterogeneity of gliomas also represents added value and, in our view, underlines the novelty of this review and points to the need for ongoing and new study concepts for the use of 5-ALA-SDT with regard to safety and effectiveness.
Round 2
Reviewer 1 Report
Comments and Suggestions for Authors
The authors have carefully and thoroughly addressed all the concerns raised in the initial review. The responses are appropriate, and the revisions made to the manuscript significantly improve its clarity, structure, and overall scientific quality. The updated version presents a more coherent and comprehensive overview of the topic. I have no further major comments and support publication in its current form.
Reviewer 2 Report
Comments and Suggestions for Authors
The authors have addressed my comments, and so, my recommendation is for acceptance.